# Genetic Variability and Management in Nero di Parma Swine Breed to Preserve Local Diversity

**DOI:** 10.3390/ani10030538

**Published:** 2020-03-24

**Authors:** Elena Mariani, Andrea Summer, Michela Ablondi, Alberto Sabbioni

**Affiliations:** Dipartimento di Scienze Medico-Veterinarie, University of Parma Via del Taglio 10, 43126 Parma, Italy; elena.mariani@unipr.it (E.M.); andrea.summer@unipr.it (A.S.); alberto.sabbioni@unipr.it (A.S.)

**Keywords:** Nero di Parma, pig, conservation, genetic diversity, pedigree, breeding strategies, inbreeding, biodiversity, autochthonous

## Abstract

**Simple Summary:**

The Nero di Parma is an Italian pig breed with a peculiar breed history. It originates from a native breed called “Nera Parmigiana”, which, in the beginning of the 20th century, was crossed with highly productive breeds, causing the extinction of the original type in the 1970s. During the 1990s a growing interest for organic products and outdoor farming brought the attention back to the local type and a breed recovery project started to reestablish the original breed. The aim of the study was to investigate the genetic diversity of the Nero di Parma breed to provide further insights for breed conservation and to propose breeding strategies.

**Abstract:**

Nero di Parma is an endangered swine breed reared in the North of Italy which nowadays counts 1603 alive pigs. The aims of this study were (i) to explore the genetic diversity of the breed at pedigree level to determine the actual genetic structure, (ii) to evaluate the effectiveness of the breeding recovery project and (iii) to potentially propose breeding strategies for the coming generations. The pedigree dataset contained 14,485 animals and was used to estimate demographic and genetic parameters. The mean equivalent complete generations was equal to 6.47 in the whole population, and it reached a mean value of 7.94 in the live animals, highlighting the quality of the available data. Average inbreeding was 0.28 in the total population, whereas it reached 0.31 in the alive animals and it decreased to 0.27 if only breeding animals were considered. The rate of inbreeding based on the individual increase in inbreeding was equal to 7%. This study showed the effectiveness of the recovery project of the breed. Nevertheless, we found that inbreeding and genetic diversity have reached alarming levels, therefore novel breeding strategies must be applied to ensure long-term survival of this breed.

## 1. Introduction

The profound changes in the production systems occurred in agriculture in the past 50 years have caused severe and undesirable impacts on the worldwide genetic variability. After the second World War, the call for animal-based products steeply increased, and rural systems were not any longer able to sustain food demand. In this context, the agricultural system has gone through a period of great changes which led to the development of intensive and commercially oriented systems. These systems necessarily promoted the use of high-yielding breeds at the expenses of the local ones. The changings, driven by the needs at that time, did not consider unfavorable side effects in the long terms. High-yielding breeds increasingly replaced traditional breeds, causing a threatening loss of genetic diversity. Swine system exactly resembles this scenario. Indeed, in the last decades, cosmopolitan lean breeds such as Landrace, Large White, or Duroc have been selected worldwide for their high productivity, at the expense of local breeds which have been crossed to improve their performance. Local breeds generally show strong adaptation to their local environment, even in the presence of harsh conditions, supporting their potential role to address the future climate changes and potential disease outbreaks. They represent a significant economic resource especially in rural areas as they have been mainly used in the context of niche markets. Lastly, conservation of small local breeds relies also on their socio-cultural value and their contribution to the development of local communities [1]. Therefore, local pig breeds and their sustainable exploitation represent a key source in the current demand for biodiversity preservation. Thus, small local breeds, even if close to extinction, need to be rescued to counteract the loss in genetic resources.

The Nero di Parma is a swine breed reared in the North of Italy, mostly in the province of Parma. Its ancestor was the “Nera Parmigiana” breed whose main characteristics were a slate colored coat, presence of wattles [2], ears hanging and facing forward, a strong conformation, exceptional resilience, an optimal adaptation to extensive conditions and excellent meat quality (Figure 1a). At the end of the 20th century, Nera Parmigiana was extinct due to the massive crossbreeding with Large White and Berkshire breeds. In the mid-90s, a growing interest arose for local breeds and their productions, usually considered qualitatively superior and for niche market. Consequently, a breed recovery project started with the aim to recover the original breed type of the Nera Parmigiana. The project initiated as part of the European Commission LEADER II initiative, which aimed to promote the development of rural areas in the late 1990s. The Local Action Group coordinated the project in synergy with the Department of Veterinary Science (University of Parma), the Italian Swine Breeders Association (ANAS) and the Regional Breeder Association (ARAER). The primary approach was indeed to perform an extensive examination of genetically connected animals that could fulfill the Nera Parmigiana original morphology. After few years of controlled matings, a slate coat nucleus was obtained. In 2006 the ANAS approved the institution of the Hybrid Breeders Book of Nero di Parma, which was then officially recognized as a breed, in 2016 (Figure 1b).

Nowadays, the live population counts 1603 pigs. In small populations the risk of permanent loss in genetic material is dramatically high, and the maintenance of genetic diversity represents one of the most important breeding goals [4]. The conservation of a breed is strictly linked to its economic relevance as it guarantees the survival of the breed in the long term. This is because the conservation of animals that are no longer economically convenient will more likely face the threat of extinction. The peculiar characteristics of the Nero di Parma breed, both in terms of adaptability to rural conditions and of meat properties, allow the breed to fit nicely into the current market demand of niche products. Actually, the intense red pigmentation and a strong marbling of the cuts of the Nero di Parma meat clearly differentiate the Nero di Parma from similar products on the market. Pedigree analysis is an effective tool to characterize the genetic diversity of a population due to its cost-effectiveness ratio, especially in the case of small populations where financial support is generally a limiting factor. Through pedigree’s information it is possible to estimate the variability and the development of the population structure in consecutive generations [5]. To assess genetic variability at genealogical level, multiple parameters can be used, such as inbreeding, average relationship, and number of founders and ancestors [6,7]. The increase of inbreeding is one of the main issues occurring in small populations, especially in the case of recovered breeds. Inbreeding can cause loss of alleles linked to economically important traits and a potential decline in performances [8]. Therefore, the implementation of breeding strategies is a key factor to avoid loss of genetic diversity and likewise to improve the economic value of the breed. The aims of this study were i) To explore the genetic diversity of the Nero di Parma breed at pedigree level in order to assess the actual genetic structure of the breed, ii) to evaluate the effectiveness of the breeding recovery project, and iii) to propose breeding strategies to preserve genetic variability in the long terms.

## 2. Materials and Methods

### 2.1. Data Avaiable, Quality Control, and Reference Populations

Since the beginning of the 2000s, ARAER (Associazione Regionale Allevatori Emilia-Romagna) in collaboration with University of Parma routinely collects identification data of Nero di Parma animals. The quality control (QC) of the data was performed checking the correctness of sows’ and boars’ registration numbers, birth date and sex identification. After the QC, the pedigree database contained 14,485 pigs (TP: total population) of which 1603 currently alive (RP: reference population): 641 males (39.99%) and 962 females (60.01%). Within RP the number of males includes the castrated males. Reference population is defined as the total number of pigs currently alive, whose pedigrees were traced back to the earliest recorded ancestors. In addition to the RP, the current breeding population (BP) was built considering alive sows and boars used for reproduction purpose. To build the BP two filters were applied: Sows and boars alive with at least one offspring (Table 1). The farm ID was used to group animals in subpopulations based on the herd of origin in order to evaluate differences in genetic diversity and breeding strategies among farms.

The quality of pedigree information was investigated using the equivalent complete generations (CGE), which is computed as follow:∑ (1/2)^n^(1)
where *n* is the number of generations between individuals and each known ancestor [9]. In addition, to evaluate the trend of the CGE throughout the studied period and among sex, we calculated CGE by birth and sex. For each animal belonging to the TP, RP, and BP (i) the number of full traced generations and (ii) the maximum number of generations traced were calculated. Generation intervals (GI) were calculated for the following pathways: Father to son, father to daughter, mother to son and mother to daughter using the average age of parents at the birth of their offspring.

### 2.2. Genetic Variability and Population Structure

The actual number of founders (f) and ancestors were calculated in the TP, RP, and BP. The effective number of founders (f_e_) and the effective number of ancestors (f_a_) were calculated according to Lacy [10] and Boichard [11] as the minimum number of founders or ancestors explaining the observed genetic diversity in the TP, RP, and BP. To evaluate whether the population had a well-balanced founder/ancestor contribution, the ratio between f_a_ and f_e_ was calculated. The ratio between f_e_ and f was computed to evaluate the presence of bottlenecks after breed foundation [12]. Founder genome equivalents (f_g_) was also computed which is defined as the number of founders that would be expected to produce the same genetic diversity if the founders were equally represented and no loss of alleles occurred [10]. Following Caballero and Toro [7], f_g_ was obtained by the inverse of twice the average coancestry of the individuals present in the reference population. The effective number of non-founders (nf_e_) was computed following Caballero and Toro [7]. Finally, the ratio between the f_g_ and f_e_ was calculated to evaluate if the population is affected by genetic drift [11]. The inbreeding coefficient (F) is the probability of an animal to be homozygous for a locus by descendant and was computed according to Meuwissen and Luo [13]. The average F was additionally computed within birth year cohorts from 2001 to 2018 to evaluate its trend during the recovery project. Moreover, the Average Relationship (AR), which is the mean relationship of each individual with the remaining population, was calculated in the TP, RP, and BP. We evaluated also the presence of mean differences in the AR among boars and sows in the BP. Average relatedness can be interpreted as the representation of the animal in the whole pedigree regardless of the knowledge of its pedigree. The individual rate of inbreeding (Δ*F*_i_) was calculated according to the approach proposed by Gutiérrez et al. [14] in the form proposed by Gutiérrez et al. [15]. This latter method allows to obtain estimate free of demographic and management effect and compute realized N_e_. The realized effective population size (N_e_) was thus estimated as proposed by Cervantes et al. [16]. The concept of N_e_ is a useful estimator of the inbreeding state in a determined population [17]. All the following analyses related to genetic variability estimations were computed in the ENDOG v4.8 software [18]. The herd ID was used to evaluate the genetic diversity within farm and to establish the genetic material exchange between farms. To evaluate the presence of different breeding strategies among herd type based on herd size, four percentile classes were calculated using the total number of registered animals. For each percentile class, the average number of pigs born in the herd, the percentage of piglets born from boars belonging to the same herd and the average CGE, F, and AR were calculated. The F statistics [19] were computed following Caballero and Toro [7] based on herd subpopulations, focusing on the current operating farms in the RP. Fixation indices (F_IS_, F_ST,_ F_IT_) were calculated to detect the reduction of heterozygosity among subpopulations based on farm ID and individuals to measure the total population differentiation. The F_ST_ values were interpreted using the qualitative guidelines proposed by Wright [19], where an F_ST_ value of 0.15–0.25 indicates large differentiation, 0.05–0.15 indicates moderate differentiation, and F_ST_ < 0.05 indicates little differentiation among populations. To evaluate the hierarchical relationship among farms in the RP, a clustering algorithm based on the F_ST_ results was used in R [20]. The R “gplots” package was used to generate a heatmap of each pairwise F_ST_ value among farms.

## 3. Results

### 3.1. Data Avaiable, Quality Control, and Reference Populations

At the beginning of 2000s, the Nero di Parma population started to grow. After the institution of the Hybrid Breeders Book of Nero di Parma in 2006, there was a steady increase in the population size. It doubled the number of individuals reaching the level of about 1000 piglets born/year in 2007, and it remained averagely stable until 2014, when a second peak occurred and 1550 newborn per year were registered. Although some fluctuations occurred, since 2014, the population is generally above the 1200 born piglets/year, except for 2018, when a reduction was registered (Figure 2). The number of breeding animals, both boars and sows, did not increase proportionally with the population. Sows were less than 5% of the total numbers of females, and boars did not reach the 1%.

The completeness of pedigree was evaluated using CGE, which was equal to 6.47 considering the TP and it reached a mean value of 7.94 and 7.22 in the RP and BP, respectively. The CGE was below 2.5 in the first six birth year (1998–2003) and it reached a mean value above 5 in 2009. Throughout the studied period a gradual increase in the CGE was found and no differences among sex were shown (Appendix A). The average number of full traced generations was equal to 3.95 in the TP and 4.85 in the RP. In the case of breeding animals (BP) the average number of full traced generations was equal to 4.52. The average maximum number of generations traced was equal to 9.31, 11.59 and 10.26 in the TP, RP, and BP, respectively. As reported in Table 2, the GI in the paternal lineages was higher than in the maternal lineages. Considering all the pathways, the average GI was equal to 2.46 years.

### 3.2. Genetic Variability and Population Structure

The main parameters that characterize the Nero di Parma genetic variability in the TP and in the subgroups RP and BP are given in Table 3. Considering TP, the actual number of founders was 32 and f_e_ was equal to 3.42. The actual number of ancestors was 16 and the f_a_ was equal to 3.0. The number of ancestors explaining the 50% of the observed genetic diversity was equal to 2 animals. The ratio between f_a_ and f_e_ was equal to 0.88 whereas the ratio between f_e_ and f was equal to 0.11. The f_g_ was equal to 2.01 and the ratio between f_g_ and f_e_ was equal to 0.59. The actual number of founders identified was 19 and 11 for the RP and BP, respectively. The number of f_a_ and f_e_ were similar for the RP and BP if compared to the TP. The average inbreeding coefficient (F) in the TP was 0.28 (SD = 0.10) with a maximum value of 0.66. In the RP the average inbreeding was equal to 0.32 (SD = 0.11) while, when considering alive breeding animals (BP) (*N* = 336), the inbreeding decreased reaching a mean value equal to 0.27 (SD = 0.09). The average AR in the TP, RP, and BP was equal to 0.50 (SD = 0.06), 0.51 (SD = 0.048) and 0.50 (SD = 0.09), respectively. In addition we evaluated AR in the BP for boars and sows separately and we found a mean value equal to 0.51 (SD = 0.09) in boars (*N* = 21) and equal to 0.50 (SD = 0.09) in sows (*N* = 315). The individual increase in inbreeding (Δ*F*_i_) was equal to 0.07, 0.06 and 0.05 in the TP, RP, and BP, respectively. In the case of realized N_e_, we found an increase in N_e_ from the TP to the BP (TP = 7.68, RP = 8.72, BP = 9.66) (Table 3).

In Table 4, the first two birth year cohorts refer to the population prior the attribution to the Hybrid Breeders Book, and the last birth year cohort refers to the population registered as a breed. Starting from early 2000s, when the breed recovery project begun, the average F gradually increased reaching a mean value of 0.31 (SD = 0.10) and a maximum of 0.62 in the last birth year cohort.

A total of 44 farms were identified in the database, of which 25 had a total of registered animals lower than 100 throughout the studied period. In Table 5 population genetic diversity parameters are presented per each herd size percentile class. The percentage of piglets born from boars belonging to the same herd varied considerably among the evaluated farms. From the percentile clustering, we observed that the percentage of internal father raised together with the herd size. We indeed found that the first percentile class (average *N* = 15) had a percentage of internal father equal to 9.31%, whereas in the fourth percentile class (average *N* = 1124) this value reached 66.70%. The major difference in terms of average inbreeding was found from the comparison of the first and last percentile, with an average F of 0.32 and 0.28, respectively.

The analysis of the RP in terms of farms available showed that only 10 farms are currently operating. The Wright F parameters F_IS_, F_ST_ and F_IT_ were calculated based on the aforementioned farms and were equal to −0.07, 0.12, and 0.06, respectively. In Figure 3 is presented the heatmap as graphical representation of the F_ST_ pairwise comparison. In the case of Farm E, the F_ST_ showed moderate to high differentiation to all the remaining farms. In contrast, seven farms showed no genetic differentiations based on the F_ST_ values.

## 4. Discussion

Nowadays, preservation of genetic diversity is crucial to overcome the dramatic loss of worldwide animal biodiversity. Biodiversity has a widely recognized role in food security and nutrition; nevertheless, our food systems is under severe threat due to the ongoing loss of genetic diversity [21]. In this context, the preservation of local pig breeds has an extreme important function. Their sustainable employment represents a reservoir of genetic variability and exerts a pivotal role in the current demand of biodiversity. In the last 20 years a great effort has been made to save biodiversity, as demonstrated by the variety of projects developed to recover native breeds as for example in Italy Cinta Senese, Casertano pig, and Mora Romagnola [22,23].

This study can be considered as the first step in deepening our knowledge on the genetic diversity of the Nero di Parma and to establish the effectiveness of the recovery project. Since the beginning of the project, the number of piglets born per year generally increased, especially after the officially recognition of the breed. This outcome suggests that the breed recovery project was effective to increase the population size, regardless the absence of economic grant from the government. Although Nero di Parma is no longer classified as at risk of extinction, the females breeding population consists of 315 females, thus it still has to be considered as endangered. Indeed, FAO [24] stated that a breed is endangered if the total number of breeding females is between 100 and 1000.

The pedigree depth is of main importance in the study of genetic diversity based on pedigree data as it affects the estimation of all the genetic parameters [8]. Thus, we evaluated the quality of the available data to ensure accuracy of our genetic parameter estimations. In our study, the mean equivalent complete generations indicates a remarkable pedigree depth, especially when compared to those published by Tang et al. [25] who found CGE lower than 3.0 in Chinese breeds farmed in Sichuan’s province. Compared to our study, Krupa et al. [26] published similar results for five different breeds of Czech Republic (CGE mean value = 7.5), whereas higher pedigree quality was found in more commercial breeds as Canadian Duroc and Lacombe, (13 and 17.5, respectively) [27]. The pedigree depth of the Nero di Parma confirmed the efficient and constant recording system in the breed which is often a limitation in the case of small populations. The generation intervals found in our study were shorter in the maternal lineages compared to the paternal ones which is in contrast to what previously found in both, local [26] and commercial breeds [25]. Nevertheless, the average GI calculated from all pathways (2.46 year) was longer than what found in the case of most commercial lines [25,26,27]. This is probably due to the extensive and rural farming system that characterized the Nero di Parma production. The latter result is in line with what is known in literature for commercial and local breeds. Local breeds have lower reproductive performances compared to commercial ones, and they are usually older at first parturition and have longer lactation period [28,29].

The ultimate objective of a recovery project is the preservation of breed genetic variability from which founders were drawn. A remarkable difference between f_e_/f was found in the TP compared to RP and BP. In particular, the f_e_/f in the TP suggested that only 11% of the initial genetic variability was kept throughout the generations. Thus, we can hypothesize that the breed experienced a loss of genetic diversity as side effect of the breeding candidates’ selection during the initial steps of the recovery project. This is a quite common scenario during a recovery project. It is known that most losses of genetic variability occur just straight after the setting up of a conservation program as likely a substantial number of founders do not give progeny to the following generations [30,31,32]. The effective number of founders and ancestors were similar to Gochu Asturcelta pig breed [32], but lower if compared to Bísaro Pig [12] and to commercial breeds used in China [25], in Czech Republic [26] or Canada [27]. A ratio between f_a_ and f_e_ close to 1 ensures the continuity of genetic origin [12]. In the Nero di Parma study, this ratio was lower than 1 if the TP is considered, but raised to 1 in the alive populations which suggests a well-balanced founder/ancestor contribution in the current population. This latter aspect highlights the effectiveness of the recovery project in terms of appropriate mating policies to conserve genetic variability, despite the initial and inevitable loss. The genetic drift, estimated as the ratio between f_g_ and f_e_, was equal to 0.59 in TP showing a clear loss of genetic diversity. This ratio was higher than in the Bunte Bentheimer (0.35) [33] but lower than in the Gochu Asturcelta pig breed (0.64) [32]. The losses of genetic variability revealed by the f_g_ being roughly 40% lower than f_a_ might be related to the partition of genetic variability among farms. We indeed found that the use of own boar within farms was high especially in the biggest farms (66.7% in the fourth percentile). Likewise, the results obtained for the F_IS_ and F_IT_ and F_ST_ highlighted the need to implement breeding strategies to assist the exchange of genetic materials among farms. Particularly, from the heatmap, seven farms showed clear genetic homogeneity, whereas in the remaining pairwise comparisons moderate to high level of differentiations were found. It is noteworthy that the largest farms were also on average the ones with the lowest inbreeding. This result is explanatory of the different breeding strategies adopted by the breeders depending on their herd size. Especially in one farm, a remarkable degree of differentiation based on F_ST_ value was found being above 0.09 in six pairwise comparisons. Interestingly, this latter farm is one of the biggest farms counting 332 currently alive animals (21% of the current alive population) of which 55 being used as breeding animals (16.4% of the current alive breeding population). This farm exhibited rather low inbreeding and coancestry compared to the remaining ones. We thus believe that, even though the genetic material exchange with other farms is not commonly practiced in this farm, yet an effective mating scheme is used within the farm. On the other hand, smaller farms showed low differentiation highlighting that the exchange of genetic material is regularly performed. Nevertheless, the remarkable high inbreeding and average relationship found in the smallest farms suggested a reduction in genetic diversity due to the extensively use of a boar family among them. Therefore, the adoption of optimal contribution selection (OCS), a method that maximizes the selection objective while constraining progeny inbreeding, might be an effective tool in this breed. The OCS might help to increase the genetic material exchange among farms in the Nero di Parma. We therefore suggest the use OCS to optimize the contribution of breeding animals and homogenized coancestry via a better exchange of genetic materials among farms. The ΔF_i_ found in the TP was equal to 7.0% which was higher than the ΔF_i_ in most commercial and local pig breeds. The ΔF_i_ was twice as high as in the Bísaro Pig [12] and consistently higher when compared to commercial breeds as Chinese Landrace, Duroc and Yorkshire [25]. In this study, N_e_ based on ΔF_i_ was equal to 7.68, 8.72, and 9.66 in TP, RP, and BP which is similar to what was found in the Gochu Asturcelta pig breed [32]. Those two populations are similar in terms of effective number of founders and ancestors even if the Nero di Parma TP is almost five-fold higher than the Gochu Asturcelta pig breed. The average inbreeding in Nero di Parma breed is higher compared to what is found in literature for more commercial herds [34]. Similar results were instead reported by Fernandes and colleagues [35] for the Portuguese Bísaro pig, which is akin to the Nero di Parma in terms of rearing system and population size. A high level of inbreeding was expected in the Nero di Parma, considering the limited number of animals from which the recovery project started. However, to keep the breed alive in the long term it is extremely important to adopt new strategies to preserve and potentially enhance the genetic diversity in the breed. In this sense, the fact that average F was lower in the breeding population compared to the whole population highlights the ongoing effort to use less related animals in the breeding program. The use of genomic tools might help in this scenario as it will lead to a complete disclosure of the genetic diversity of the Nero di Parma. Moreover, genomic data might be also used to disentangle its ancestry in the context of Mediterranean pig breeds [1]. The Nero di Parma farmers are still using natural mating. The implementation of artificial insemination (AI) together with OCS could be an effective tool (i) to increase the exchange of breeding males between farms and (ii) to create specific mating schemes to control and manage relationships within and among farms. Those two latter aspects are especially relevant in the Nero di Parma as different mating strategies based on farm size were found in this study which led to diverse results in terms of average inbreeding and coancestry. Nevertheless, AI must be used with caution since it potentially decreases the number of breeding males. Since breed conservation is highly linked to its economic value, we ultimately suggest developing specific DNA markers for the Nero di Parma to increase its traceability. This latter aspect is especially important as high quality products are more vulnerable to frauds that cost to Italian food industry billion of euros per year [36]. The effectiveness of those novel molecular tools has been already proved in mono-breed products such as the Parmigiano Reggiano cheese from Reggiana breed or traditional pork cured products from Cinta Senese and Nero Siciliano breeds [37,38,39]. Therefore, genomic traceability will allow protection of Nero di Parma production as well as the farmers, the consumers and the breed itself.

## 5. Conclusions

To conclude, in this study we unraveled the genetic diversity of the Nero di Parma at pedigree level and we highlighted few main future strategies that might be implemented for its conservation. The results of our study showed that the recovery project of the Nero di Parma was effective despite the absence of economic national support. To increase the effective population size and lower the rate of inbreeding, we suggest the use of OCS together with AI which can in turn effectively help to improve the exchange of genetic materials among farms. A genomic implementation of this work is needed to completely disclose the state of genetic diversity in Nero di Parma breed and to be able to protect its high-quality production.

## Figures and Tables

**Figure 1 animals-10-00538-f001:**
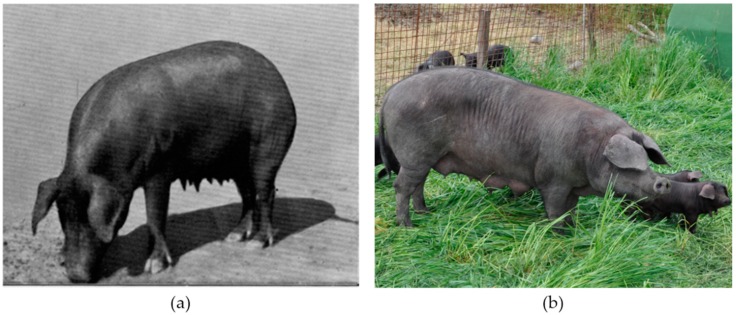
(**a**) An example of the original type called “Nera Parmigiana” found in an historical book [3]; (**b**) Nero di Parma sow with piglets (Authors’ own repository).

**Figure 2 animals-10-00538-f002:**
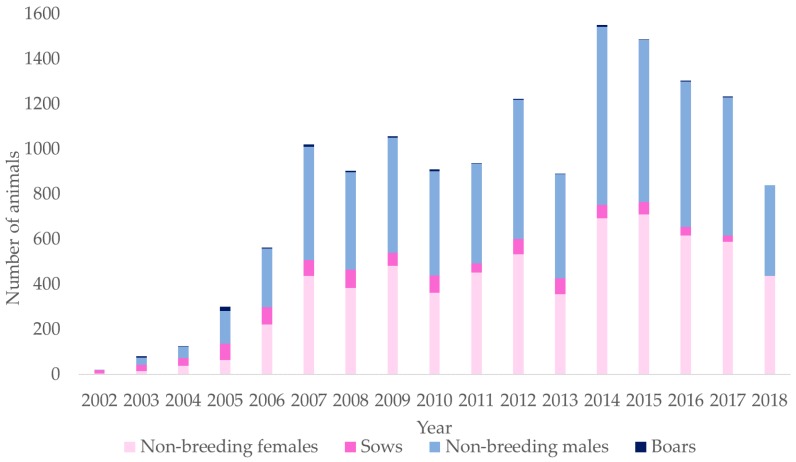
Animals registered per year of birth divided by sex and with or without progeny: Boars, non-breeding males, sows, and non-breeding females.

**Figure 3 animals-10-00538-f003:**
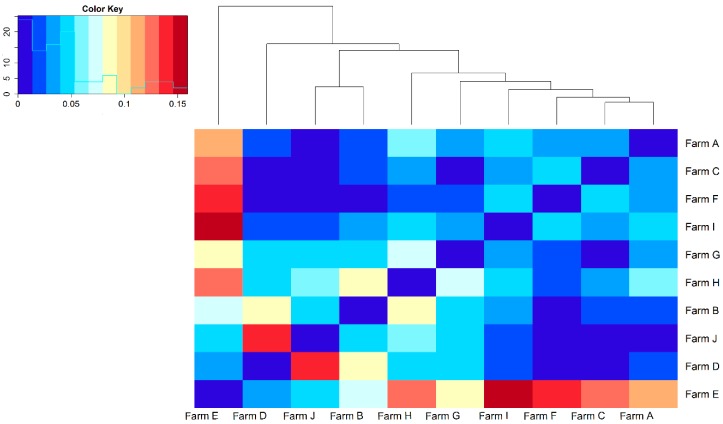
Heatmap of F_ST_ results for the 10 farms found in the reference population (RP). The cells in the plot are colored by Fixation index values with deeper colors indicating higher (red) or lower (blue) genetic distances.

**Table 1 animals-10-00538-t001:** Description of the data available in the entire pedigree dataset (TP), in the reference population (RP) and in the breeding population (BP).

Parameters	TP ^1^	RP ^2^	BP ^3^
Numbers of pigs	14,485	1603	336
Numbers of males	7197	641	21
Numbers of females	7288	962	315
Numbers of pigs with no progeny	13,507	1267	0

^1^ TP = total population, total number of animals present in the database; ^2^ RP = reference population, number of pigs currently alive; ^3^ BP = breeding population, breeding animals in the reference population.

**Table 2 animals-10-00538-t002:** Generation Interval (GI) calculated for each parental pathway.

Pathway	*N* ^1^	GI	SD ^2^
Father-Son	96	2.65	1.71
Father-Daughter	876	2.69	1.32
Mother-Son	94	2.26	1.08
Mother-Daughter	870	2.24	0.98
Total	1936	2.46	1.20

^1^*N* = number of individuals contributing to GI within the pathway; ^2^ SD = Standard Deviation.

**Table 3 animals-10-00538-t003:** Main genealogical parameters for the Nero di Parma breed in the total, reference, and breeding populations (TP, RP, and BP, respectively).

Genealogical Parameter	TP	RP	BP
Population size	14,485	1,603	336
Number of Founders (f)	32.0	19.0	11.0
Number of Ancestors	16.0	13.0	11.0
Equivalent complete generations (CGE)	6.47	7.94	7.22
Average inbreeding (F)	0.28	0.32	0.27
Average Relationship (AR)	0.50	0.51	0.50
Effective Number of Founders (f_e_)	3.42	3.00	3.00
Effective Number of Ancestors (f_a_)	3.00	3.00	3.00
Founder genome equivalents (f_g_)	2.01	1.81	1.91
Effective Number of non-founders (nf_e_)	4.88	4.58	5.29
f_e_/f	0.11	0.16	0.27
f_a_/f_e_	0.88	1.00	1.00
f_g_/f_e_	0.59	0.60	0.64
Individual increase in inbreeding (Δ*F*_i_)	0.07	0.06	0.05
Realized effective population size (N_e_)	7.68	8.72	9.66

**Table 4 animals-10-00538-t004:** Inbreeding coefficient (F) in the total population (TP) e in the breeding animal within TP from the start of breed recovery project to 2018.

Birth Year Cohort	Total Population (TP)				Breeding Animal within TP			
	*N* ^1^	F ^2^	SD ^3^	Highest F	*N*	F	SD	Highest F
2001–2003	130	0.18	0.16	0.50	72	0.18	0.16	0.59
2004–2006	991	0.31	0.13	0.61	208	0.29	0.13	0.62
2007–2009 *	2983	0.29	0.12	0.66	233	0.28	0.15	0.66
2010–2012	3072	0.28	0.10	0.61	200	0.28	0.10	0.47
2013–2015	3930	0.26	0.07	0.53	192	0.27	0.10	0.53
2016–2018 **	3379	0.31	0.10	0.62	73	0.29	0.09	0.55
Grand Total	14,485	0.28	0.10	0.66	978	0.27	0.13	0.66

^1^*N* = number of pigs in the cohort. ^2^ F = Inbreeding coefficient, expressed as mean values in the year cohort. ^3^ SD = standard deviation, * Officially recognized as Hybrid breeding pig, ** Officially recognized as a pig breed.

**Table 5 animals-10-00538-t005:** Population structure, average inbreeding (F) and relationship coefficients (AR) among animals born in different herd size percentile class.

Herd Class ^1^	Farms ^2^	*N* ^3^	CGE	Own Father (%) ^4^	F			AR		
					Mean	SD ^5^	Highest ^6^	Mean	SD ^5^	Highest ^6^
1	11	15.0	6.6	9.31	0.32	0.15	0.57	0.50	0.10	0.61
2	11	42	5.9	11.69	0.30	0.10	0.58	0.52	0.05	0.64
3	11	130	6.7	11.75	0.31	0.11	0.61	0.49	0.07	0.60
4	11	1124	6.5	66.70	0.28	0.10	0.66	0.50	0.06	0.63

^1^ Four percentile classes based on the herd size were calculated using the total number of registered animals; ^2^ Farms = number of farms in each herd class; ^3^
*N* = average number of pigs born in the herd; ^4^ Own Father = percentage of piglets born from boars belonging to the same herd; ^5^ SD = standard deviation; ^6^ Highest = highest inbreeding (F) and average relationship (AR) coefficient within herd percentile class.

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
