# Peer review of "Genetic Variability and Management in Nero di Parma Swine Breed to Preserve Local Diversity"

_animals, 2020, doi:10.3390/ani10030538_

Round 1
Reviewer 1 Report
Major comments
The manuscript deals with an important and interesting topic, which is about estimating genetic diversity parameters for an Italian local pig breed. With a few breeds playing a major role in the industry, preserving local genetic resources is important for the future. It is interesting the overview about the farms and the practical solution authors give with the genetic material exchange.
Nevertheless, some errors are reported in the manuscript, i.e. at line 257 the results are inverted compared to the section “Results”, and at line 288 the percentage written is 90%, instead of 66.7%. Also, in “References” section there are some mistakes, e.g. L8, L20 and L22. Please, revise this section. In materials and methods, the other two parameters linked to completeness of pedigree are cited but seems to be mixed. Parameters are: (i) the number of full traced generations; (ii) the maximum number of generations traced and (iii) the equivalent complete generations. So, please, clarify this aspect and report the results if you cited these parameters in M&M.
It might also be desirable to discuss not only FST value, but also FIS and FIT results, as well as to better define the four percentile classes, reporting the number of farms which direct in each percentile and also the range of consistency analyzed.
Minor comments
L13: Please correct the dates format.
L13: Change “biological” to “organic”.
L19: Please, highlight the three aims of the study, for examples using i), ii), iii); in addition, write “to evaluate”.
L22: Change the term to “equivalent complete generations” as Gutiérrez and Goyache (2005) reported in “A note on ENDOG: a computer program for analyzing pedigree information”. Please, revise it in the whole manuscript (e.g. L244).
L36: Change to “worldwide genetic variability”.
L46: Are you speaking about “The global strategy for the management of farm animal genetic resource or Global Plan of Action for Animal Genetic Resources?” However, include year and citation.
L70: Replace “that” with “which”.
L71: Add a comma between “breed” and “in 2016”.
L81: Substitute “factors” with “parameters”.
L84: I suggest eliminating “from few animals only”, the concept is just expressed by “recovered”.
L87: Change to “likewise to improve”
L88: Add a comma before “are key factors”.
L90: Correct with “to propose” and “performances”.
L102: Please rephrase the sentence, e.g. “current breeding population (BP), which was built taking into account sows…”.
L104: Please find a synonymous for “herd of belonging” or rephrase the sentence, it is confused. Groups are the 10 farms or are based on herd origins?
L109: Maintain the three classes of population analyzed written in the same way, namely Total Population or total population etc., in the whole manuscript.
L111: Change to “equivalent complete generations”.
L112: Add the citation.
L122: I suggest substituting “unity” with 1.
L126: Change “subject” to “individual”.
L131-135: Please add the citation for all formulas you consider.
L141: Correct in “average inbreeding and relationship coefficients”.
L148: Rephrase in “The R gplots package”.
L149-150: Please, move the sentence to L135.
L159: Replace “with the exception” with “except”.
L163: I suggest rephrasing the variables in the graph: the four classes should be all as singular or all plural terms, in addition, “no progeny” is not nice to read.
L167: There is a double space, please remove it.
L168: Remove “whereas”, there is no contrast.
L174: The same, change to “individuals”.
L178: Not use “whereas” if you have not a contrast! In this case you have two results which you don’t compare each other.
L179: Put “considering the TP” as the first part of the sentence.
L184: Change to “attribution to the Hybrid Breeders Book”.
L193: Move “yet” after the verb.
L198-199: I would remove the point, leaving for example “average N=15”.
L198: Add a comma after the percentage.
L203: Change to “average inbreeding (F) and relationship coefficients (AR)”.
L204: Remove percentages from the table, you wrote it in the column title. As before N instead of N.
L207: I would add “herd percentile class” to the table description.
L219: Change “has” to “have”.
L223: Add “s” to the verb.
L228: Remove “s” to the verb.
L230: Remove “s” to the verb.
L233: I think “which” is more appropriate.
L242-244: this is a very important concept, so, please add citation.
L246: Correct with “Sichuan’s province”.
L256: Add a comma after “indeed”.
L259: Krupa et al. (2015) reported opposite results than yours. Please correct this part.
L275: I invite authors to make comparison of the rate of inbreeding value obtained for this local breed, with what could find in literature. Could be interesting know the rate of inbreeding of commercial breeds, for example.
L281: Change to “to improve”.
L308: Change “lower” to “to decrease”.
L365: Please check carefully this citation. The last two authors do not exist and the information about the type of publication is missing (it is a presentation to a Symposium held in Turin).
Author Response
Dear reviewer, thank you very much for your overall positive evaluation and for your useful comments. Please find attached our answers.

Reviewer 2 Report
The paper form Mariani et al copes with genetic management and conservation of an Italian local pig breed.
Biodiversity and conservation are hot topics and it is always interesting having studies and researches from different breeds and countries.
I have some comments/suggestions for the authors
A general comment regards the lack of details of the recovery project. I think that some additional information should be included in the introduction session. Did the recovery project include any breeding policies?
In the discussion session I think that the authors should cite some previous projects focused on pig conservation (e.g. Cinta Senese http://om.ciheam.org/om/pdf/a41/00600110.pdf), especially regarding the policies applied to manage conservation.
Line 12: unfortunately ; I would not use such an adverb. This is a personal opinion not originated from actual facts. Probably, when the decision was taken it seemed to be the most appropriate
Line 35-42 I do agree that past decisions have caused current problems but I think that we should be a little bit more objective. What I mean is that the breeding decisions taken in the past were based on particular needs. So the “profound changes” you are talking about shouldn’t be considered in a negative way. They did what they thought it was good. I would like you to mitigate a little bit your very first statement. Possibly saying that past decisions, driven by the needs at that time, did not consider unfavorable side effects
Line 53-54 “They represent a significant economic resource especially in rural area as they have been mainly used in the context of niche markets.” This is partly or mainly due to the financial support given by regional governments. This is still the main problem in managing endangered species: how to find the right economic balance? Do you have any comments about it?
Line 111-113/153-154 Did you calculate CGE by year of birth and sex? It would be interesting to include some infos (possibily a graph) about it
Line 138: “…using the total number of registered animals.”
Line 190: why didn’t you simply plot the average Inbreeding by year of birth? It would easier to visualize the difference between TP and BP population. Did you estimate the average relationship between boars?
Line 193: I think that it would be better and more informative to include the number of herd per each percentile. You have only 4 percentiles so it is quite straightforward
Line 195-198. Those results, I think, are quite expected because very small farmers do prefer to get males from outside, not only because of management reason but also because in very small populations there is always a “main breeder” who acts like a leader. Which is the pedigree quality across herd size? Did you calculate CGE by herd size? It could be interesting to look at this statistics. Line 199, why not fitting a regression using all Herd data (possibly weighed by the number of records)? My feeling is that using such a grouping might shrinkage your results. Moreover, I guess you have quite a right skew distribution when looking at herd size and using percentiles in such a situation might be a little bit inappropriate.
Line 203. I think that you should use “herd size class” in place of “percentile class of consistency” “Population structure, inbreeding coefficient (F) and average relationship (AR) among animals born in different herd size class”. You can then add a footnote clarifying how each class has been created.
Line 201. Are there only 10 active herds? This is not very clear. Try to clarify
Line 212. This statement should go before line 210 “In the case of Farm E…”
Line 213. “…massive and indiscriminate…” Here again I think that both adjectives are too strong and suggest a negative idea. For sure the loss of biodiversity is a problem but we can’t jeopardize previous ‘selection’ choices, they were based on particular needs. We are now aware of the problems that such choices have involuntarily created and we have to figure out a way to recover such a situation.
Line 215. The figure title is not stand-alone. The reader have to look back in the text to know what RP means.
Line 281-283. I think that we should focus on less related breeding animals and not on less inbred individuals. Obviously the two parameters are connected but I’m interested in how much a boar is related to a particular farm and not on how much he is inbred.
Line 290-291. The largest farms are often the ones with less accurate pedigree. That’s the reason why it would be interesting to look at pedigree quality at herd level (comment on line 195-198).
Line 306-307. “We suggest enhancing the breeding animals’ contribution in the next generations to guarantee the future survival of the breed” I don’t completely understand what you are suggesting, could you try to reformulate it?
Line 307-309 “More effort…” I think that simply applying the OCS would give you amazing results. You have a good pedigree structure and tools like EVA (https://www.nordgen.org/en/farm-animals/resources/ocs/) can be easily implemented
Author Response
Dear referee, thank you very much for your positive overall evaluation and thank you for your useful comments. Please find attached our answers.

Reviewer 3 Report
Manuscript ID: animals-736394 titled “Genetic variability and management in Nero di Parma swine breed to preserve local diversity” by Elena Mariani et al.
Pedigree information of the Nero di Parma pig breed was analysed to explore the breed’s genetic diversity and to propose new breeding strategies to improve (?) the genetic variability and the performance of the breed.
This kind of research is particularly interesting to me.
However, unfortunately, in its present form the manuscript presented lacks of general interest, has several technical weaknesses and do not present new strategies to preserve the genetic variability of this valuable genetic resource.
The latter issue is very important. Most Discussion (L278 onwards) is very general, merging very different possibilities that are not linked to the results presented. Do the authors pretend to implement a selection programme with limits to the increase in inbreeding? Would the authors like to use IA to unbalance the contributions of the founders to the present population and equalise the founder’s representation? Do the authors implement a long-term programme to keep balanced the contributions of the founders as proposed in Goyache et al., 2003, JABG, 120, 95-103? In this respect, it is worth mentioning that two recent papers dealing with the use of genealogical information to preserve pig genetic resources are missing:
- Menedez et al., 2016, Czech Journal of Animal Science, 61, 140-143
- Paixao et al., 2018, Czech Journal of Animal Science, 63, 452-461
The weaknesses of the research presented are of major importance as well:
- Authors defined two reference populations (RP and BP) but results are only referred to one of them (BP; Table 3).
- No information on the actual number of founder for TP, RP and BP and, therefore, an interested reader cannot know if fe = 32 is high or low. Ratio fa/fe is 0.5. What are the causes of this value? When and how the bottleneck occurred? Why authors did not compute genome equivalents to summarise all losses of genetic variability due to drift?
- Authors computed deltaF defining the “last” generation. This cannot be done in “real world” populations (in which generations overlap is the rule) but in a very general way. There is consensus that in livestock populations the best way to compute deltaF is the “individual increase in inbreeding” (and its counterpart “individual increase in coancestry”) on the individuals belonging to a predefined reference population. This allows to obtain estimate free of demographic and management effect and compute realized Ne. Related to that the discussion on the FAO’s criteria does not make much sense after Leroy et al., 2013, Genetics Selection Evolution 2013, 45:1. The comments on this paper, freely available at https://gsejournal.biomedcentral.com/articles/10.1186/1297-9686-45-1/comments, can give the authors light on the concerns summarised above.
Author Response
Dear referee, thank you very much for your feedback.
Please find attached our answers to your comments.

Round 2
Reviewer 2 Report
I would like to thank the authors for accepting most of the suggestions. You did a good job
Author Response
Dear Referee,
Thank you very much for your insightful comments and for your positive evaluation of our work.
Reviewer 3 Report
Manuscript ID: animals-736394.V2 titled “Genetic variability and management in Nero di Parma swine breed to preserve local diversity” by Elena Mariani et al.
This is a new version of a previously revised paper.
The manuscript has been considerably improved. However, the work still needs major improvement before considering it for publications in Animals.
My main concern is that the effectiveness of the breeding recovery project (L19-20) has not been actually assessed. This is the new information of general interest I’m still missing.
Authors justify the need of taking measures to preserve genetic variability on the fact that the ratio fe/f is very low (L293 and thereafter). However, this is a common scenario: most losses of genetic variability occur just after the start of the recovery programme. See as a classical example Alvarez et al., 2008, Relationship between genealogical and microsatellite information characterising losses of genetic variability: empirical evidence from the rare Xalda sheep breed. Livestock Science, 115, 80-88. doi:10.1016/j.livsci.2007.06.009, and many others later. Interesting enough, the fact that ratio fa/fe is roughly 1. This is quite surprising and question the interpretation of the results presented. In “standard” scenarios fg < fa < fe <f (please refer ratio as fa/fe to give an accurate idea of the losses of genetic variability). If ratio fa/fe = 1, no additional bottlenecks occurred after the implementation of the mating policy due to the abusive use of a low number of breeding animals, suggesting that breeders have applied appropriate mating policies to conserve genetic variability (see Menendez et al., referenced in the text). However, there are losses of genetic variability due to the fact that fg is about 40% lower than fa (or fe in this particular case). Authors have not identified the causes of these losses. Is this the partition of genetic variability among farms? Most farms have low differentiation (Fig, 3). One of them may have F value exceeding coancestry. Is this due to a particular founder effect in this farm due to the almost exclusive use of a boar family? Can the coancestry be homogenized at a population level via the use of bars purchased in other farms or using the boars of this farm in other farms? This discussion should be connected with the demographic analysis (Table 5). In its present form the discussion on Table 5 has not a clear meaning.
Related to that, the other major objective of the research presented “to propose breeding strategies for the coming generations.” (L20), has not been achieved neither. Authors simply state that “Moreover, the results obtained for the FIS and FIT and FST highlighted the need to implement breeding strategies to assist the exchange of genetic materials among farms.” (L323-325) with no proposal of specific measures. Moreover, the proposal of an OCS program will fit to a desired future scenario (and even could be supported by the results obtained in the work if correctly presented) but does not derive from the results of the manucript.
Minor concerns
A major part of the Introduction section (L35-63) consists on a justification of the need of preserving genetic stocks. This is assumed by most readers of Animals. The Introduction section can be considerably shortened with no losses of information. This will allow that an interested reader is not distracted from the main message of the paper.
L412-413: does this paper exist? Please revise the References section.
Author Response
Dear referee,
Thank you very much for your insightful comments. Please find attached our point by point responses.
